# Using quantitative methods to understand leaf epidermal development

Chi Kuan, Shao-Li Yang, and Chin-Min Kimmy Ho

Institute of Plant and Microbial Biology, Academia Sinica, Taipei City, Taiwan

leaf epidermal development; quantitative methods; SLGC; stomata; cell state.

**Author for correspondence:**
C.-M. K. Ho,
E-mail: chmho@gate.sinica.edu.tw

### Abstract

As the interface between plants and the environment, the leaf epidermis provides the first layer of protection against drought, ultraviolet light, and pathogen attack. This cell layer comprises highly coordinated and specialised cells such as stomata, pavement cells and trichomes. While much has been learned from the genetic dissection of stomatal, trichome and pavement cell formation, emerging methods in quantitative measurements that monitor cellular or tissue dynamics will allow us to further investigate cell state transitions and fate determination in leaf epidermal development. In this review, we introduce the formation of epidermal cell types in Arabidopsis and provide examples of quantitative tools to describe phenotypes in leaf research. We further focus on cellular factors involved in triggering cell fates and their quantitative measurements in mechanistic studies and biological patterning. A comprehensive understanding of how a functional leaf epidermis develops will advance the breeding of crops with improved stress tolerance.

## 1. The epidermis as a load-bearing layer for organ morphogenesis

The 'epidermal-growth-control' or 'tensile skin' theory was proposed over a century ago based on the experimental result that the removal of the outer tissue (peripheral cell layers) resulted in a faster elongation of the inner stem tissue than with intact stems (Kutschera & Nikas, 2007). This observation suggested that the epidermis restricts organ growth. The molecular mechanism was then validated using epidermally expressed Brassinosteroids (BRs) signalling to promote cell expansion and rescue the dwarfism of *br* mutants (Savaldi-Goldstein et al., 2007). The load-bearing epidermis is also required for organ integrity. A lack of coordination among the layers of the epidermis leads to cracks in tissues, but this phenotype can be fixed by reinforcing epidermal cell wall stiffness (Asaoka et al., 2021; Verger et al., 2018). Recent studies and reviews have elucidated how mechanical forces shape plant organs and determined how tensile stress influences cell fate determination (Bidhendi et al., 2019; Malivert et al., 2018; Sampathkumar et al., 2014; Trinh et al., 2021; Yang et al., 2022). Here, we focus on the molecular mechanisms underlying epidermal fate determination and patterning using quantitative measurements at the tissue or cellular level.

## 2. Cell types of the leaf epidermis

The leaf epidermis in Arabidopsis comprises three cell types: trichomes, pavement cells, and stomata. Trichomes are hair-like structures on the leaf surface that offer a defensive barrier against herbivores and insects (Handley et al., 2005). So-called pavement cells 'pave' the leaf surface, forming another physical barrier that prevents attacks by pathogens such as bacteria and fungi. Additionally, pavement cells are covered by a cuticle on their external side (facing the air) to reduce water loss through evaporation (Lu et al., 2012). Arabidopsis stomata are valve-like pores on the leaf surface that are formed by a pair of kidney-shaped guard cells. They permit leaves to take up carbon dioxide from the atmosphere while allowing water evaporation, which drives water uptake from the roots. These three cell types coordinate their development and placement to form proper compartments to keep carbon dioxide for mesophyll cells to generate sucrose-based energy. Therefore, the leaf epidermis provides a barrier that protects mesophyll cells and is important for balancing water evaporation and carbon dioxide acquisition.

The precursor cells that form the leaf epidermis are derived from the outermost layer of the shoot apical meristem (SAM). The transcription factor WUSCHEL (WUS), whose encoding gene is expressed in the organising centre, interacts with CLAVATA1 (CLV1), CLV2 and CLV3 at the SAM (Clark et al., 1993, 1995; Fletcher et al., 1999; Kayes & Clark, 1998; Mayer et al., 1998). WUS and CLV establish a feedback loop that controls the size of the SAM and maintains the organising centre (Schoof et al., 2000). The central zone further differentiates into the L1 and L2 layers, with epidermal cells differentiating from L1, and meso-phyll cells and fundamental stem tissues differentiating from L2. The homeobox-containing transcription factors ARABIDOPSIS THALIANA MERISTEM LAYER1 (ATML1) and PROTODER-MAL FACTOR2 (PDF2) regulate the establishment of L1 (Abe et al., 2003). In Arabidopsis (*Arabidopsis thaliana*) leaves, the L1 layer forms the protoderm, which differentiates into trichomes, stomata, or puzzle piece–like pavement cells on the adaxial side of leaves or into stomata or pavement cells on the abaxial side (Figure 1a).

## 2.1. Trichome development

Epidermal development is spatially and temporally regulated by cell fate factors, cell polarity and inhibitory signalling. For example, the development of initial trichome cells is controlled by GLABRA2 (GL2), a class IV homeodomain-containing basic leucine zipper (HD-bZIP) transcription factor (Rerie et al., 1994). GL2 promotes trichome maturation through cell expansion, branching, and cell wall establishment (Szymanski et al., 1998). *GL2* expression is activated by a complex, termed the MBW complex, which consists of the R2R3 MYB protein GL1, the basic helix-loop-helix (bHLH) transcription factor GL3, and the WD40-repeat protein TRANS-PARENT TESTA GLABRA1 (TTG1) (Payne et al., 2000). The MBW complex activates *GL2* transcription to induce trichome cell fate and promotes the translocation of the R3 MYB protein TRIP-TYCHON (TRY) to neighbouring non-hair cells, where it replaces GL1 in the MBW complex (Schnittger et al., 1999; Zhao et al., 2008). The MBW complex containing TRY represses *GL2* expression and inhibits trichome initiation in non-hair cells (Schellmann et al., 2002; Schnittger et al., 1999; Figure 1b). In addition to this activator–inhibitor model, trichome patterning also requires an activator–depletion mechanism that traps TTG1 in trichome initial cells to enhance trichome production (Balkunde et al., 2020; Digiuni et al., 2008; Pesch & Hulskamp, 2009), which we discuss in more detail in Section 4.2.

## 2.2. Stomatal development

Stomatal development is driven by a transcription factor cas-cade starting with the related bHLH proteins SPEECHLESS (SPCH), MUTE, and FAMA (MacAlister et al., 2007; Ohashi-Ito & Bergmann, 2006; Pillitteri et al., 2007). SPCH promotes asymmetric cell division in the protoderm, resulting in a small cell (meristemoid) and a large cell (stomatal lineage ground cell, SLGC) (MacAlister et al., 2007; Pillitteri et al., 2007). MUTE commits the meristemoid to the stomatal fate and activates the expression of genes involved in symmetric division and cell cycle regulation (Han et al., 2018; 2022; Pillitteri et al., 2007). FAMA then drives symmetric cell division to generate a pair of guard cells (Ohashi-Ito & Bergmann, 2006). bHLH transcription factors typically associate as homodimers or heterodimers. SCREAM (SCRM) and SCRM2 function together with SPCH, MUTE and FAMA along this cell lineage to drive stomatal production (Kanaoka et al., 2008).

Stomatal distribution follows the one-cell-spacing rule; two stomata never directly touch each other and are thus separated by at least one cell. Cell-to-cell communication initiates from the meristemoid, which secretes the peptides EPIDERMAL PATTERNING FACTOR1 (EPF1) and EPF2. These peptides then bind to receptor protein kinases from the ERECTA family (ERf) and to the coreceptor TOO MANY MOUTHS (TMM). This receptor-coreceptor complex then activates a MITOGEN-ACTIVATED PROTEIN KINASE (MAPK) cascade that inhibits SPCH activity in SLGCs (Bergmann et al., 2004; Hara et al., 2007; 2009; Ho et al., 2016; Hunt & Gray, 2009; Lampard et al., 2008; Lee et al., 2012; Shpak et al., 2005). Polar proteins located in SLGCs, including BREAKING OF ASYMMETRY IN THE STOMATAL LINEAGE (BASL), members of the BREVIS RADIX family (BRXf), and POLAR (Dong et al., 2009; Houbaert et al., 2018; Pillitteri et al., 2011; Rowe et al., 2019), form a complex on the opposite side of the meristemoid and serve as a scaffold for recruiting the inhibitory signalling components MAPK3 and MAPK6 to phosphorylate and inhibit SPCH, thus preventing SLGCs from adopting a stomatal fate (Lampard et al., 2008; Zhang et al., 2015; Figure 1c). The resulting SLGCs can either become pavement cells or reinitiate several rounds of asymmetric divisions to produce a range of patterns on the epidermis (Gong, Alassimone, et al., 2021a; Ho et al., 2021).

## 2.3. Pavement cell formation

Cells other than trichomes or stomatal initial cells differentiate into pavement cells. No cell-type-specific transcription factor has yet been described to be associated with pavement cell formation. Arabidopsis pavement cells take on their intriguing puzzle shape via the action of auxin gradients, cytoskeletal function, epidermal tension from cell proliferation and expansion, and the force from tissue growth (Grones et al., 2020; Sapala et al., 2018; Xu et al., 2010; 2014). Pavement cell interdigitation is abolished in mutants lacking auxin biosynthesis function or the auxin efflux transporter PIN-FORMED1 (PIN1) (Xu et al., 2010). Auxin actives two Rho GTPases that promote the formation of the complementary lobes and indentations. The GTPase RHO OF PLANTS2 (ROP2) and ROP-INTERACTIVE CRIB MOTIF-CONTAINING PROTEIN4 (RIC4) activate F-actin formation for the protrusion of lobes, while ROP6 and RIC1 promote microtubule organisation for indenta-tions (Fu et al., 2009). The bridge between auxin and the action of Rho GTPase signalling is AUXIN BINDING PROTEIN1 (ABP1), which interacts with the plasma membrane–localised receptor-like transmembrane kinases (TMKs), leading to the activation of Rho GTPases and the formation of the puzzle shapes characteristic of pavement cells on the surface of Arabidopsis leaves (Xu et al., 2014; Figure 1d).

The cell cycle provides an additional layer of control in epider-mal cell fate determination and maintenance. For example, the for-mation of pavement cells and trichomes is usually associated with increased ploidy; however, stomatal cells remain diploid (Larson-Rabin et al., 2009; Melaragno et al., 1993; Xu et al., 2014; Zuch et al., 2022). The mitotic cell cycle comprises G1, S, G2, and M phases and results in two diploid (2n) daughter cells; endoreplication, a specialised type of cell cycle, increases ploidy (4n, 8n, 16n, or more) by undergoing DNA replication by skipping cell division. Endoreplication is crucial for trichome fate maintenance (Bram-siepe et al., 2010). When CYCLIN-DEPENDENT KINASE A;1 (CDKA;1) is mutated or when the CDK inhibitor *KIP-RELATED PROTEIN1* (*KRP1*, also named *INHIBITOR/INTERACTOR WITH CYCLIN-DEPENDENT KINASE1* [*ICK1*]) is ectopically and stably

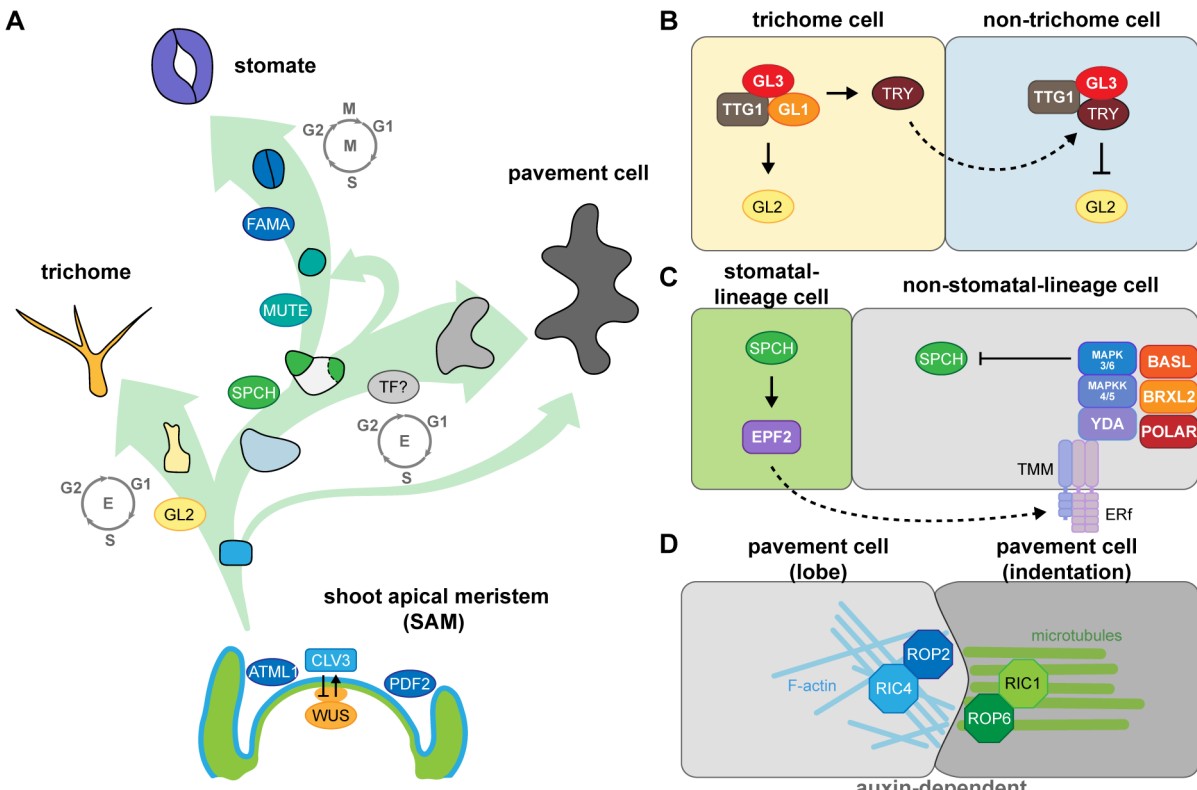

**Fig. 1.** Key factors involved in fate determination and leaf epidermal patterning during development. (A) Three trajectories describe the three distinct cell types of the leaf epidermis. The development and size of the shoot apical meristem (SAM) is controlled by the CLAVATA3 (CLV3)-WUSCHEL (WUS)-mediated negative feedback loop. The transcription factor (TF) genes *ARABIDOPSIS THALIANA MERISTEM LAYER1* (*ATML1*) and *PROTODERMAL FACTOR2* (*PDF2*) are expressed in the outermost layer (L1, blue) in the SAM to establish epidermal cell identity. Cells in the L1 layer called protoderms give rise to trichome-initiating cells (yellow), meristemoid mother cells (light blue), and pavement cells (grey). During trichome development, the major TF, GLABRA2 (GL2), drives cells to become polyploid trichome cells; the differentiation state is maintained by endoreduplication. To form a stomate, the master TF, SPEECHLESS (SPCH), initiates asymmetric cell division in a meristemoid mother cell to generate a small meristemoid (green) and a stomatal lineage ground cell (SLGC, white). MUTE then replaces SPCH and commits the cell to becoming a guard mother cell. Subsequently, FAMA in the guard mother cell drives symmetric cell division, resulting in a pair of guard cells. An SLGC can either undergo differentiation to become a pavement cell or divide asymmetrically again to produce another stomate (labelled by a dashed line). To date, no cell-type-specific driving factors have been associated with pavement cell formation. However, pavement cell maturation is often coupled with endoreduplication. (B) The cell–cell communication between trichome and non-trichrome cells relies on TRIPTYCHON (TRY). Before trichome initiation, GL1, GL3, and TRANSPARENT TESTA GLABRA1 (TTG1) form a stable MBW complex to activate *GL2* transcription. GL2 accumulation promotes trichome cell fate. Besides GL2, the MBW complex also activates *TRY* expression in trichome cells. TRY tends to move to the neighbouring non-trichome cell to replace GL1 and disrupt the formation of the MBW complex, thus repressing *GL2* and inhibiting the trichome cell fate. (C) Stomatal patterning follows the one-cell-spacing rule, meaning that two stomata never directly contact each other, and this signalling is mediated through peptide-mediated inhibitory signals and polarity establishment during stomatal development. SPCH drives the accumulation of a signalling peptide, EPIDERMAL PATTERNING FACTOR2 (EPF2), in meristemoids. The secreted peptides then bind to members of the receptor-like kinase ERECTA family and the receptor-like protein TMM on the surface of neighbouring cells. The binding triggers a MAPK cascade that phosphorylates and inhibits SPCH activity, thus preventing stomatal fate in these neighbouring cells. The interplay of chemical transduction and the polarity complex consisting of BASL, BREVIS RADIX-LIKE2 (BRXL2), and POLAR is required for asymmetric cell division and serves as a scaffold for recruiting and exerting inhibitory signalling in non-stomatal lineage cells. (D) Pavement cell formation is controlled biochemically by auxin. The high auxin concentration at the lobe initiation site induces the asymmetric accumulation of RHO-RELATED PROTEIN FROM PLANTS (ROP)-ROP-INTERACTIVE CRIB MOTIF-CONTAINING PROTEIN (RIC) in the two pavement cells. For the lobing cell (left), the recruitment of ROP2 and RIC4 helps the formation of actin filaments (F-actin) and further results in the protrusion of the cell. For the indented cell (right), ROP6 and RIC1 stabilise microtubule organisation under parallel direction and further pull the cell, forming an indentation.

expressed, trichomes only undergo mitotic division; these plants have fewer trichomes and these trichomes have fewer branches (Bramsiepe et al., 2010).

In the following sections, we summarise quantitative methods for characterising leaves, trichomes, stomata, and pavement cells as well as protein polarity and cell cycle progression (Table 1). By obtaining quantitative data, we can then trace the transitions between cell states and start to understand cell decisions over the course of development.

## 3. Quantitative methods to study leaves

### 3.1. Leaf shape

Arabidopsis forms elliptical true leaves with a serrated edge. Classical parameters such as leaf area, length (height), width, and circu-larity are used to describe leaf shapes. The ratio between the length and width determines whether the leaf is more circular or elliptical. Leaf circularity reflects the degree of serration of the leaf margin (Figure 2a). The program LAMINA (Leaf shApe deterMINAtion) was developed in 2008 to extract such leaf measurements from scanned leaf images and performs well on both single and compound leaves (Bylesjö et al., 2008). LAMINA can be employed to determine the number of leaf serrations and boundary coordinates, which describe the degree of asymmetry in leaf shape. However, LAMINA cannot detect petioles or describe highly complex leaves. In Arabidopsis, there is no clear boundary between the leaf blade and petiole, which may hinder the quantification of overall leaf area (which would include the petiole here). Using the program MorphoLeaf, the petiole can be removed manually to increase the accuracy of leaf shape quantification (Biot et al., 2016). Furthermore,

**Table 1.** Quantitative tools to measure leaf epidermis features

| Tools | Category | Purpose | Description | Reference |
|---|---|---|---|---|
| Cell-type features characterisation | | | | |
| MorphoLeaf | Leaf shape | Extracts leaf contour, leaf sinus, and leaf tips from multiple leaf images. Determines hierarchization of leaf teeth. Provides quantitative leaf shape parameters | A plug-in running on the Free-D software | Biot et al., 2016 |
| LeafI (Leaf Interrogator) | Leaf shape | Quantifies leaf shape and leaf rosette from leaf image. Extracts leaf contour and leaf blade. Performs shape-space analysis and data visualisation | A GUI-based pipeline implemented in Python 3.5 with a PyQt5-based GUI | Zhang et al., 2020 |
| LIMANI (Leaf Image Analysis Interface) | Leaf vein | Extracts the leaf vascular network by automatic image segmentation. Measures venation patterns | A web-based application with the 'grey-scale mathematical morphology image analysis algorithm' | Dhondt et al., 2012 |
| LAMINA (Leaf Shape Determination) | Leaf shape | Quantifies and extracts the leaf area and leaf shape from images of diverse plant species. | A graphical application implemented in Java. | Bylesjö et al., 2008 |
| PaCeQuant | Pavement cell | Automatically segments images and quantitatively analyzes pavement cell shape characteristics | A plug-in of ImageJ | Möller et al., 2017 |
| GraVis (Visibility Graph) | Pavement cell | Describes pavement cell shape with visibility graph. Quantifies pavement cell protrusion and indentation. Analyzes characteristics of pavement cell shape | A GUI-based pipeline implemented in Python 3 | Nowak et al., 2021 |
| Stomata Counter | Stomata | Automatically identifies and counts stomata from microscopy images via deep learning | A convolutional neural network system implemented in Python 2.7 | Fetter et al., 2019 |
| LeafNet | Pavement cell and stomata | Segments and quantifies stomata and pavement cells from brightfield microscopy images with a hierarchical deep learning technique | A Python-based package that can also be used online | Li et al., 2022 |
| MorphoGraphX MGX2.0 | Organ and tissue | Visualises and analyzes 4D biological confocal images. Extracts cell geometry data and organ/cell shape parameters | A software written in C++ and developed on GNU/Linux | de Reuille et al., 2015 Strauss et al., 2022 |
| PlantSeg | Plant organ, tissue, and cell | Provides 3D segmentation of plant tissues into cells with deep learning | A convolutional neural network system implemented in Python | Wolny et al., 2020 |
| Cell fate regulators measurement | | | | |
| POME (Polarity Measurement) | Polarity | Quantifies the characteristics of cell polarity with a semi-automated pipeline | A pipeline composed of Fiji macro and R scripts | Gong et al., 2021a; 2021b |
| Cytrap (Cell Cycle Tracking in Plant Cell) | Cell cycle | Monitors the cell cycle progression in Arabidopsis. | *HTR2pro:CDT1a-RFP* for S+G2 and *CYCB1pro:CYCB1-GFP* for late G2+M. | Yin et al., 2014 |
| PlaCCI (Plant Cell Cycle Indicator) | Cell cycle | Monitors the cell cycle progression in Arabidopsis | *CDT1apro:CDT1a-eCFP* for G1, *HTR13pro:HTR13-mCherry* for entire cell cycle, and *CYCB1;1pro:CYCB1;1-YFP* for late G2+M | Desvoyes et al., 2020 |
| SPACE (Stomata patterning autocorrelation on epidermis) | Peptide and stomata pattern | Quantitatively determines how signalling peptides influence stomatal patterning | A Python script with spatial autocorrelation algorithm | Zeng et al., 2020 |
| DII-Venus | Auxin response | Examines auxin response level. | Venus fluorescent protein fused with Aux/IAA auxin-interaction domain (DII). | Brunoud et al., 2012 |
| R2D2 (Ratiometric DIIs) | Auxin response | Examines auxin responses. | Ratiometric analysis of DII-Venus and mDII-Venus | Liao et al., 2015 |
| AuxSen, Auxin Biosensor | Auxin level | Quantifies *in vivo* auxin levels and visualises auxin distribution. | A FRET-based biosensor containing an engineered tryptophan repressor from *E. coli* | Herud-Sikimić et al., 2021 |

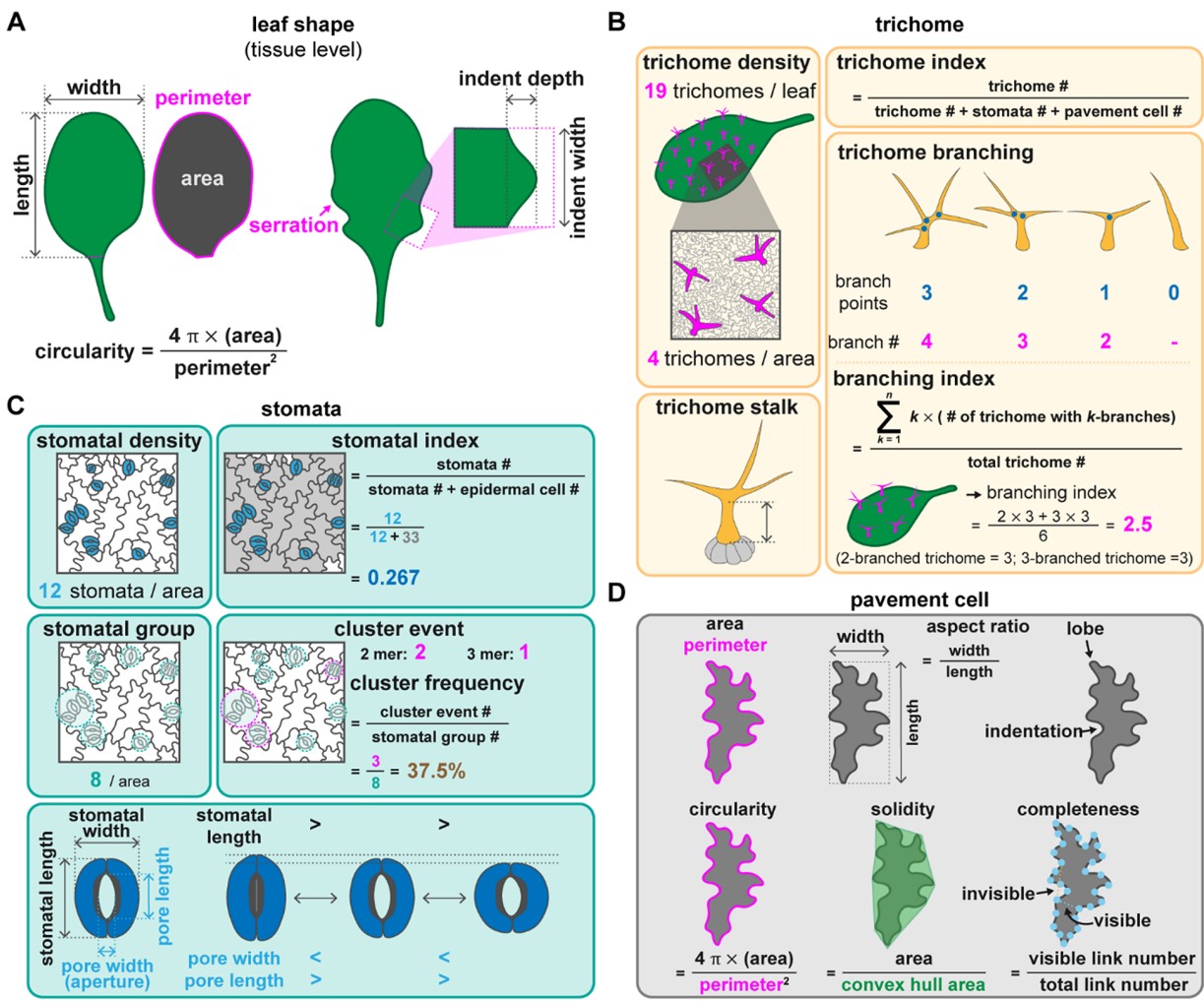

**Fig. 2.** Quantitative methods to describe leaf morphology and the different types of epidermal cells. (A) Leaf shape can be described by its length, width, and the length/width ratio. Circularity calculated by ($4\pi\times$leaf area)/(square perimeter) indicates that the leaf shape is approximately round (circularity = 1) or contains serrations (circularity closer to 0). If serrations are visible on a leaf, the indent length and width can be used to describe the extent of serration. (B) Trichome phenotypes can be measured by the number of trichomes on a leaf and trichome morphology. Trichome density can be scored over an entire leaf or a given area. Trichome index is the number of trichomes divided by the number of total epidermal cells including trichome, stomatal, and pavement cells over a given area. The trichome stalk, the branching number, and the branching points can be used to represent trichome shape. The trichome branching index represents the average number of branches in a trichome population. (C) Stomatal phenotypes can be described by stomatal number, pattern, and dynamics. Stomatal density represents the number of stomata over a given area. The stomatal index is defined as the number of stomata divided by the total number of epidermal cells. Stomatal groups are the number of stomatal islands in a given area. Adjacent stomata are defined as one group. A cluster event represents the number of islands with adjacent stomata. Cluster frequency represents the error rate of forming stomatal clusters. The movements of stomata can be described by stomatal length and width. Pore length and width excluding guard cells may also be used. During stomatal opening, the stomatal and pore lengths decrease, while the pore width increases. (D) Pavement cell phenotypes can be described according to their shape including the parameters of area, perimeter, length and width. The aspect ratio calculates the ratio between cell length and width. The jigsaw-puzzle shape can be described by the number of lobes and indentations, which can be expressed as circularity, solidity, and completeness. Circularity is calculated as ($4\pi\times$cell area)/(square perimeter). Solidity is the ratio between cell area and convex hull area. Completeness is the ratio between visible and total possible links. A visible link represents the link between two nodes without crossing over the cell outline. An invisible link is defined as a link between two nodes that crosses over the cell outline. All three parameters have maxima equal to 1. They decrease when the cell takes on the shape of a jigsaw-puzzle piece.

landmark-guided reparameterization in MorphoLeaf enables the determination of the average leaf shape based on several biological samples. This feature is useful when tracing leaf developmental trajectories and obtaining morphometrics. Many classical parameters can be collected and analysed in a principal component analysis (PCA) to investigate the main differences between plant species, varieties, or accessions. The GUI (graphical user interface)–based package LeafI (Leaf Interrogator) in the Python environment can also be used to analyse leaf images and generate datasets in which classical leaf shape parameters are assembled (Zhang et al., 2020). Moreover, LeafI provides statistical analysis and visualisation tools to facilitate and accelerate analysis. The PCA function of LeafI

can clearly detect differences in leaf shape between mutants. For example, an analysis of single and higher-order mutants of the *WUSCHEL-RELATED HOMEOBOX* (*WOX*) family, the latter of which have spindle-like leaves, indicated that WOX proteins are required for lateral leaf expansion (Zhang et al., 2020).

Besides the classical leaf shape parameters, the program LIMANI (Leaf IMage ANalysis Interface) provides information about the vascular patterns of leaves (Dhondt et al., 2012). Using ethanol-cleared leaves and darkfield images to increase image contrast is critical for analyses via LIMANI, which can score vascular density, the number of branching points, and the number of lamina and areola. Combined with classical parameters, measuring

these additional parameters may improve our understanding of the relationship between vein and shape developments.

Because Arabidopsis leaves are flat, most programs focus on two-dimensional (2D) leaf structures. Remmler and Rolland-Lagan (2012) computationally quantified the three-dimensional (3D) structure (curvature) of the leaf adaxial surface using brightfield and fluorescence z-stack images (Remmler & Rolland-Lagan, 2012). In a study of the development of the first true leaf in Arabidopsis, the 3D structure of the leaf changed due to spatial heterogeneity in growth, even when the classical leaf shape parameters did not appear to change significantly (Rolland-Lagan et al., 2014). This model may elucidate the relationship between growth and shape and the effects of mechanical properties on the leaf surface.

## 3.2. Trichomes

In Arabidopsis, trichomes are present on the adaxial side of true leaves but not on cotyledons. Mutations in genes associated with trichome development influence the number or presence of trichomes (Payne et al., 2000; Rerie et al., 1994). Trichomes with fewer branches are observed in plants with mutations in genes associated with cytoskeleton organisation or endoreplication (Roodbarkelari et al., 2010; Tian et al., 2015). Their individual shape and their spatial organisation can be described using different sets of parameters. The cell proportion is usually calculated as trichome density (number per leaf or per given area). However, given that bigger cells can result in lower cell counts, the trichome index can be used to eliminate the size effect, which is the number of trichomes divided by the total number of epidermal cells (trichome + stomatal + pavement cells) (Figure 2b).

Trichome morphology can be described by the properties of trichome stalks and branches. Stalk length and diameter can be extracted from trichome images taken with a scanning electron microscope (Szymanski et al., 1999). Zhang and Oppenheimer (2004) developed a simple method to isolate trichomes from leaves and observe them under light microscopy. Using the definition of stalk length as the distance from the basal cells to the first branching point and branch length as the distance between branching points, Zhang et al. (2005) reported that a mutation in *IRREGULAR TRICHOME BRANCH* (*ITB*) resulted in a cell expansion defect with no effect on the branching pattern. Branching can be described as the number of branches and branching points for a mature trichome (Figure 2b). Typically, Arabidopsis trichomes have three branches with two branching points (Abe et al., 2004). If branching is highly variable between individual cells, the branching index can be used as a metric, which is calculated as the total number of branches ($k$) multiplied by the number of trichomes with $k$ branches and divided by the total number of trichomes (Figure 2b; Vadde et al., 2018). The branching index thus represents the average number of branch points in a trichome.

Trichome patterning is a critical character to describe the epidermis. Clusters of trichomes occur in mutants defective in endoreplication, such as the *siamese* (*sim*) mutants (Walker et al., 2000), or mutants in repressors of trichome development, such as *try* mutants (Schnittger et al., 1999). Except for strongly abnormal phenotypes, such as contiguous trichome clustering, uneven distributions of trichomes along the leaf surface can be described. The point-pattern approximation that we refer uses the nearest neighbour distance (NND), where each individual trichome/stomate is considered as a point (Clark & Evans, 1954). Okamoto et al. (2020) used the point–pattern approximation and

a GL3pro:GL3-GFP reporter to monitor gene expression under control and heat treatments. They observed that the trichome patterning gene *GL3* is expressed heterogeneously. This finding suggested that the noise resulting from either the abundance of cellular components or environmental conditions may lead to variable patterns in a biological system (Okamoto et al., 2020). Therefore, to predict the regulatory pathways underlying the formation of a given pattern, this variability needs to be considered. Greese et al. (2014) reviewed several methods using the point pattern approximation. By comparing the experimental data and simulation of activator mobility in trichome initiation, they found that with the increasing activator mobility, the local noise decreases and the trichome pattern becomes more regular, illustrating less variability. It suggests that even a highly regulated biological process can be substantially affected by variability, which may explain the known high flexibility of plant development.

## 3.3. Stomata

Stomatal distribution is an example of biological patterning. To describe the stomatal patterns (clustered, even, or random) in different species, the point–pattern approximation similar to the described analysis in trichomes is often used. However, in the case of stomata on the surface of the leaf epidermis (shown in Figure 2), the size of one stomate limits the placing of neighbouring cells. The disk–NND (disk–null model) method considers the influence of stomatal size (Naulin et al., 2017). Results from the disk–null model are indeed more accurate than a simple NND approach in defining stomatal clusters in Arabidopsis, which has a high stomatal density (Naulin et al., 2017).

Epidermal patterning can be visualised by staining with propidium iodide or using plasma membrane markers to delineate the outline of epidermal cells. Yang et al. (2022) applied five measurements to systematically describe stomata: stomatal density, stomatal index, stomatal group, cluster event, and cluster frequency. Stomatal density is the number of stomata in a given area. To remove the effect of the expansion of pavement cells from both stomatal and non-stomatal lineages, the stomatal index represents the number of stomata as a proportion of the total number of epidermal cells. The stomatal density and index are influenced by the number of initial stomatal cells and adjacent stomatal clusters, which are derived from the same origin of asymmetrically divided sister cells (a single stomatal group). Therefore, the number of stomatal groups reflects the initial number of stomatal formation events. A stomatal cluster event is the number of stomatal clusters with more than two adjacent stomata. The cluster frequency is the ratio between the number of stomatal cluster events and stomatal groups and indicates the error rate of cluster formation (Figure 2c). To describe the stomatal arrangement, stomatal evenness using the method of minimum spanning tree, stomatal divergence using the method of distance to the gravity and stomatal aggregation using NND were proposed (Liu et al., 2021). Based on the simulation results, the stomatal divergence is influenced by the stomatal number while stomatal evenness and stomatal aggregation are more consistent in the stomatal number ranging from 15 to 50 (Liu et al., 2021). Since the placement of stomata can be more regular in monocot than in dicot plants, plant species should be considered when making a comparison. The stomatal production and patterning are influenced by the environment, therefore, using the aforementioned methods to study the patterns of stomatal distribution may reflect their development, adaptation, and even evolution.

Stomatal conductance is a measure of the degree of stomatal opening and can be monitored as a sum of stomatal responses in a leaf using a photosynthesis system (Li-Cor system). Therefore, to directly probe changes in stomatal apertures, opening and closing events are measured at the cellular level under a microscope. Because Arabidopsis stomata are approximately elliptical, they can be described by the length and width or the area of their aperture (Figure 2c). Changes in stomatal size or stomatal aperture under various environmental conditions can be measured to investigate the response speed of stomata (Nagatoshi et al., 2016; Tsai et al., 2022).

### 3.4. Pavement cells

Pavement cells make up the bulk of the leaf surface and interlock with each other like a jigsaw puzzle. This pattern is created and maintained by an auxin-mediated pathway and the mechanical stress between epidermal cells (Sapala et al., 2018; Xu et al., 2010). The interdigitation of the lobes and indentations provides balance between the growth direction of two neighbouring cells (Sapala et al., 2018). Modulating cell wall stiffness also plays a role in shaping pavement cells through controlling pectin composition (Altartouri et al., 2019; Haas et al., 2020). Therefore, the morphology of pavement cells may serve as a proxy for the mechanical properties of the leaf epidermis. Several parameters have been used to describe pavement cells, including the aspect ratio, the number of lobes and indentations, circularity, solidity, and convexity (Figure 2d). PaCeQuant, LobeFinder and LOCO-EFA all use a boundary-based approach to detect the lobes (Möller et al., 2017; Sanchez-Corrales et al., 2018; Wu et al., 2016). Among them, PaCeQuant, an ImageJ-based tool, provides 27 pavement cell-shape features (Möller et al., 2017). PaCeQuant either automatically detects cell boundaries of pavement cells from a confocal image or uses an input file with segmented cells. In PaCeQuant, circularity is defined as $4\pi$ multiplied by the area of a pavement cell divided by the squared perimeter (Figure 2d). Convexity is defined as the perimeter of the convex hull divided by the perimeter of the cell. Solidity is defined as the area of the convex hull divided by the area of the cell (Möller et al., 2017). These three parameters represent the lobing degree of a cell. They can be used together with pavement cell area, perimeter, and length/width ratio to generate a phylogeny map based on pavement cell shapes (Vofely et al., 2019).

GraVis is a Python 3.0 tool that uses a visibility map (Figure 2d) in a network-based approach to describe pavement cells (Nowak et al., 2021). The outline of a pavement cell is equally marked with several nodes (Figure 2d). A link is visible when it connects two nodes without crossing the cell outline, while invisible links cross the cell outline. A local maximum (more visible links) indicates indentation, and a local minimum (fewer links) indicates a lobe. A visibility matrix (the correlation between every node, including their link lengths) and completeness (the ratio between all possible links and the actual number of links) can be used to group similar shapes together and describe the lobing degree of a pavement cell (Nowak et al., 2021).

### 3.5. Large-scale, time-lapse tracking

Quantitative cell imaging provides unique spatial and temporal information at the cellular level in a multicellular context; however, it requires a huge amount of analytic power. A trend of developing computational tools to efficiently measure and extract valuable information on biological phenotypes and dynamics is

emerging. Some tools used for the quantification of leaf epidermal images are listed in this section. StomataCounter was developed to automatically count stomata (Fetter et al., 2019). The input files can be images from differential interference contrast microscopy and brightfield observations. This tool is useful for field-collected specimens in which no fluorescent dye can be deployed easily to mark the cell outline. Because the method requires high-quality images, those below a given quality cut-off need to be eliminated to obtain reliable results.

To measure and quantify the properties of every cell in a growing tissue, the accurate segmentation of individual cells from volumetric images is required. Therefore, leaves stained with propidium iodide or transgenic plants expressing a fluorescent plasma membrane marker are usually used to increase the contrast of images. The image segmentation tools Cellpose (Stringer et al., 2021) and PlantSeg allow performing 3D cell segmentation and extract quantitative measures at the single cell level (Wolny et al., 2020). Because a stomate is composed of two guard cells, the two guard cells are considered individual cells rather than a single stomate when working on cellular segmentation. To solve this problem, LeafNet applies a hierarchical strategy to first identify stomata and then segment pavement cells using stomata-masked images (Li et al., 2022). Therefore, LeafNet can be used to obtain cell counts and sizes of individual stomata as well as pavement cells in a single analysis of a two-dimensional image.

MorphoGraphX (MGX) is a platform that analyzes morphogenesis from biological images (de Reuille et al., 2015; Strauss et al., 2022). The recent MGX2.0 version implements the local coordinate system to a growing tissue or organ (Strauss et al., 2022). By adding such spatial information, the measurements of cell size and shape as well as gene expression at single time points or cell proliferation and growth rates over time allow quantification in a living organism and comparison between cell types. This advanced implementation has facilitated the identification of growth differences between leaves and sepals. In sepals, growth is more distal in the early stages and then moves towards the base as the sepal develops (Strauss et al., 2022). By contrast, the proliferation and expansion zones are relatively fixed in a proximal-distal fashion along the leaf length during leaf development (Fox et al., 2018). Since cells with a common biological function tend to have similar geometric, positional, and expression attributes, the integration of positional information and shape morphology can substantially aid in assigning cell fate (Strauss et al., 2022). For example, the coordinate system in a radially symmetric tissue makes it possible to distinguish between the epidermis, the cortex, and the stele in a root. The Arabidopsis gynoecium consists of the replum and lateral valves tissue, which has stomata (Strauss et al., 2022). Therefore, the organ coordinates can first help identify a region of interest and then use cell geometry to distinguish stomata from other elongated cells (Strauss et al., 2022). Moreover, advanced geometric analysis in time-lapse data such as lineage tracking, growth analysis, and cell division analysis in MGX2.0 will improve our understanding of spatiotemporal dynamics of cellular behaviours in the context of a developing organ.

## 4. Factors associated with cell fate determination and epidermal patterning

Cell geometry and dynamics such as polarity, cell cycle, cell size, cell division, and intercellular signalling such as cell-to-cell communication and auxin are known to play a role in cell fate deter-

mination and ultimately lead to the establishment of a biological pattern. Here, we introduce their functions in plant epidermal fate acquisition with a focus on quantitative measurements and the use of such quantitative data in understanding epidermal development.

## 4.1. Polarity

Spatiotemporal control is important for triggering different cell fates in asymmetrically dividing sister cells. During stomatal development, a polarly localised protein is required in SLGCs after asymmetric cell division for the MAPK-mediated repression of SPCH activity and the inhibition of stomatal formation (Dong et al., 2009; Zhang et al., 2015). Loss of function in polarly localised proteins such as BASL or its interacting proteins belonging to the BRX family results in stomatal clusters (Dong et al., 2009; Rowe et al., 2019). Cortical-localised BASL was shown to form a scaffold that recruits the MAPK KINASE KINASE (MAPKKK) YODA and MPK3/6 to activate signalling at the cortex (Zhang et al., 2015). The activation of MAP kinase signalling at the polarised site, therefore, reinforces the feedback loop between BASL polarisation at the cortex and the inhibition of SPCH in the nucleus (Zhang et al., 2015). BR signalling is also a regulator of stomatal production, although the underlying mechanism is more complex (Gudesblat et al., 2012; Kim et al., 2012). BR acts as a positive regulator by inhibiting the MAPK cascade, leading to the over-accumulation of SPCH and stomatal formation (Kim et al., 2012). However, BR can also be a negative regulator by directly phosphorylating and destabilising SPCH (Gudesblat et al., 2012). This dual role was solved by looking at the subcellular distribution of BRASSI-NOSTEROID INSENSITIVE2 (BIN2), a GLYCOGEN SYNTHASE KINASE3 (GSK3)/SHAGGY-like kinase, before and after asymmetric cell division (Houbaert et al., 2018). BIN2 suppresses BR signalling by relocating from the cytosol to the nucleus to inhibit BR downstream transcription factors (He et al., 2002; Kim et al., 2009). One of the scaffold proteins, POLAR (Pillitteri et al., 2011), spatially modulates the nucleo-cytoplasmic partitioning of BIN2 in the stomatal lineage. Before asymmetric division, BIN2 interacts with the polarity complex at the cortex. While this interaction is disrupted after asymmetric cell division, BIN2 relocates to the nucleus to destabilise SPCH in SLGCs, leading to asymmetric cell fates (Guo & Dong, 2019; Houbaert et al., 2018). Besides BIN2, recent work on the BRI1 SUPPRESSOR1-LIKE (BSL) family of protein phosphatases has also revealed that the spatial modulation of MAPK signalling at the subcellar level is key for asymmetric fate acquisition (Guo et al., 2021; 2022).

Plant cell polarity and tissue mechanics is an emerging field that needs to be taken into account for both biochemical and mechanical signalling in plant development (for reviews please see (Gorelova et al., 2021; Ramalho et al., 2022). The polarity degree of the crescent length normalised to cell perimeter can be used to quantify the crescent size (Gong, Varnau, et al., 2021b; Zhang et al., 2015; Figure 3a). A smaller crescent size reflects a lower inhibitory effect on SPCH, leading to over-proliferation of stomata (Yang et al., 2022; Zhang et al., 2015). Misoriented polarity can also disrupt stomatal patterning (Yang et al., 2022). Indeed, the degree of polarity of the polar protein BRX-like2 (BRXL2) in the leaf epidermis varies depending on its tissue location (Bringmann & Bergmann, 2017). At the leaf base, BRXL2 tends to face towards the midrib in both the left and right halves of the leaf; therefore, the angle between the midrib and the direction of the BRXL2 crescent can be used to measure the tissue-wide orientation of polarity proteins (Bringmann & Bergmann, 2017; Yang et al., 2022;

Figure 3a). The migration of nuclei that is polarity-driven and mediated by microtubules and actin is also important for setting up asymmetric cell division (Muroyama et al., 2020).

Post-embryonic patterning is a dynamic process that depends on cell rearrangements, polarity, asymmetric division, and growth. The information collected from time-lapse imaging to trace the expression of SPCH and the dynamic location of the polarity protein BASL was assembled into a model that revealed a postmitotic polarity-switch mechanism that governs plant stem cells to generate their neighbours while spacing themselves apart (Robinson et al., 2011).

## 4.2. Cell-to-cell communication

Trichome distribution is mediated by the intercellular mobility of several patterning proteins. Many inhibitors, such as TRY, CAPRICE (CPC), and ENHANCER OF TRY AND CPC (ETC), can move between cells (Digiuni et al., 2008; Kurata et al., 2005; Wester et al., 2009; Zhao et al., 2008). Among the activators (TTG1, GL1, and GL3), only TTG1 can move from cell to cell (Bouyer et al., 2008). TTG1 was shown to be depleted around trichomes in wild-type plants but not in gl3 mutants, suggesting that GL3 traps TTG1 in trichomes, where its encoding gene is strongly expressed (Bouyer et al., 2008). Two models have been proposed to explain trichome patterning: the activator–inhibitor model and the activator–depletion model (Pesch & Hulskamp, 2009). The activator–inhibitor model posits that the activator complex promotes the production of inhibitors that move to the neighbouring cells, where they repress trichome formation, as shown in Figure 1b. The activator–depletion model assumes the trapping of mobile TTG1 by GL3 in incipient trichomes, thus enhancing trichome production (Digiuni et al., 2008; Pesch & Hulskamp, 2009). The complexity of protein interactions and intercellular mobility among trichome patterning proteins makes it challenging to analyse the relative contribution between these two models. However, using weak ttg1 alleles to understand the regulatory circuits of trichome patterning indicated that TTG1 severs a key component in both the activator–inhibitor and activator–depletion models (Balkunde et al., 2020). The authors further determined that the core trichome patterning modules in both models differ in their activation of the long-range inhibitor CPC and the short-range inhibitor TRY (Balkunde et al., 2020).

In contrast to the two models of trichome development, stomatal patterning relies on cell-to-cell signalling through secreted peptides. The inhibitory signalling peptide family EPF (Hara et al., 2007; 2009; Hunt & Gray, 2009) and the activator peptide Stomagen (Sugano et al., 2010) synergistically influence stomatal patterning. Using mosaic analysis and the computational pipeline SPACE (stomata patterning autocorrelation on epidermis), a recent study showed that both EPF1 and Stomagen have non-cell-autonomous effects, with EPF1 having a broader effective range (100–150 mm) than Stomagen (~60 mm) (Zeng et al., 2020). This spatial correlation with the effective range of small peptides provides information at a longer-range distance as opposed to the local information about the pattern.

The interplay of cell-to-cell signalling together with the regulatory circuit that controls stomatal initiation is the key to generating a pattern, in addition to the polarity-switch mechanism already mentioned. In a two-dimensional spatial patterning model of the stomatal lineage, the SCRM-SPCH complex forms a positive feedback loop that promotes the stomatal fate as well as the production of the secreted peptide EPF2 and the receptor modifier TMM,

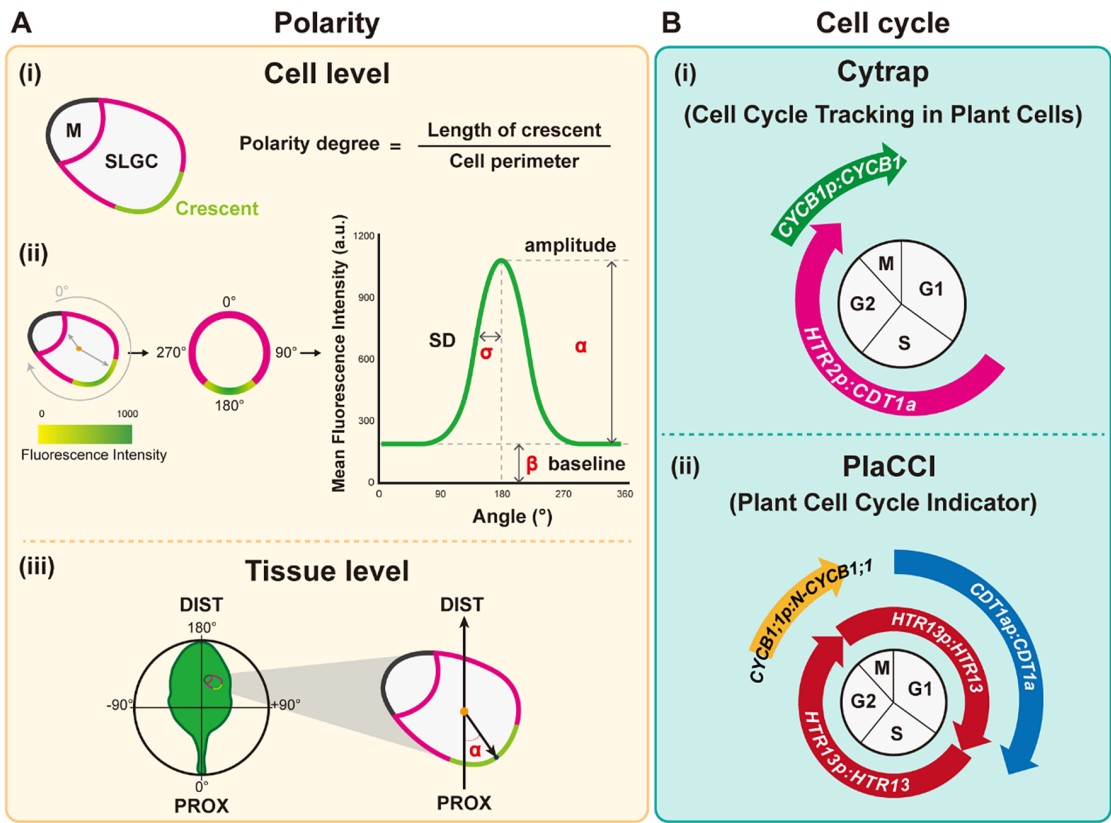

**Fig. 3.** Quantitative tools for measuring polarity and the cell cycle. (A) Polarity measurements. (i) The polarity degree is calculated by the crescent length (green) relative to the cell perimeter (green+pink). M: meristemoid. SLGC, stomatal lineage ground cell. (ii) Workflow for Polarity Measurement (POME), a Fiji-based semi-automated pipeline for polarity quantification: First, a line of 0° to 180° angle is defined by the centroid to the centre of polar protein mass. Second, the cell outline is reconstructed and visualised by quantification of the fluorescence intensity. Third, standard deviation (SD,$\sigma$), amplitude ($\alpha$), and baseline intensity ($\beta$) derived from the fluorescence intensity through Gaussian fitting quantify the degree of polarisation in the plasma membrane, with higher SD values representing higher polarity levels. (iii) Tissue-wide polarity orientation can be indicated by the angle ($\alpha$) between the leaf midrib and the connection between the cell centroid (orange dot) and the midpoint of the polarity crescent. (B) Cell cycle measurements. (i) Cytrap (Cell cycle tracking in plant cells) line. The dual-colour cell cycle reporter line uses *HTR2pro:CDT1a-RFP* and *CYCB1pro:CYCB1-GFP* to indicate S/G2 and G2/M, respectively. (ii) PlaCCI (Plant Cell Cycle Indicator) line. The three-colour cell cycle reporter line adopted *CDT1apro:CDT1a-CFP*, *CYCB1:1pro:CYCB1;1-YFP*, and *HTR13pro:HTR13-mCherry* to indicate G1, G2/M, and the entire cell cycle, respectively.

which in turn inhibits SPCH and SCRM. Thus, the resulting negative feedback loop reinforces the formation of a stomate according to the one-cell-spacing rule (Horst et al., 2015). This work provided an example using an intrinsic molecular framework to generate a self-organising two-dimensional patterning of stomatal linage initiation (Horst et al., 2015). After stomatal fate is determined, stomatal cells will undergo one round of symmetric cell division to produce a pair of guard cells. How do cells divide only once? Downstream targets of MUTE, consisting of a cell cycle regulator and its transcriptional repressor, were proposed to form a typical incoherent type I feed-forward loop to orchestrate and generate this single symmetric division event (Han et al., 2018).

### 4.3. Auxin

Auxin is a highly diffusible phytohormone that plays a crucial role in plant morphogenesis and growth by responding to intrinsic and external cues. The spatiotemporal distribution of auxin modulates the patterning and placement of cells and organs. Mathematical modelling has been used to precisely describe how auxin coordinates plant growth (Allen & Ptashnyk, 2020; Smith et al., 2006; van Berkel et al., 2013). Since auxin gradients drive root architecture and tropic responses, previous studies have investigated the link between auxin and root development. Through cell-type-specific induction of auxin biosynthesis and morphokinetic analysis, Hu

et al. (2021) quantitatively analysed root growth and skewing and created a toolbox for studying auxin-mediated plant development at a single-cell level.

Auxin is also an important factor controlling leaf development. Pavement cells are often used to investigate non-transcriptional ABP1-TMK-mediated auxin signalling (Xu et al., 2014). Auxin initiates interdigitated growth in leaf pavement cells by controlling TMK1 and ROP signalling. TMK1 also facilitates the formation of flotillin1-associated ordered nanodomains at the plasma membrane, manages cortical microtubule organisation, and changes the shape of pavement cells (Pan et al., 2020).

A survey of auxin dynamics during stomatal development established that auxin levels decreased in the small sister cell (meristemoid) after asymmetric cell division (Le et al., 2014). In one study, a fluctuating auxin gradient was observed in spirally formed pavement cells and was associated with the jigsaw-puzzle shape of pavement cells on the epidermis (Grones et al., 2020). This study used a synthetic auxin output reporter, *DR5:Venus-NLS* (with the *DR5* promoter driving the expression of an expression cassette encoding the fluorescent protein Venus fused to a nuclear localization sequence [NLS]), to trace SLGC divisions. The authors detected an association between increased auxin output and spirals of SLGC divisions. Moreover, the outermost SLGC with a high auxin level tended to become a lobed cell (Grones et al., 2020). Because auxin orchestrates a mitotic-to-endocycle switch in Ara-

bidopsis (Ishida et al., 2010), auxin may determine the transition from SLGCs to pavement cells.

Two additional auxin sensors, R2D2 (ratiometric version of two DIIs) (Brunoud et al., 2012; Liao et al., 2015) and the auxin biosensor AuxSen (Herud-Sikimic et al., 2021), have been deployed to monitor auxin responses (R2D2) and concentrations (AuxSen) during development. The DII domain is the auxin-dependent degradation domain II of an Aux/IAA protein, so the fluorescence signal of DII-Venus decreases when auxin levels are high. A mutated DII domain (mDII) that is insensitive to auxin-mediated degradation can be used as a control for semi-quantitative measurements, whereby the mDII-ntdTomato/DII-Venus ratio is proportional to the levels of auxin signal transduction. The auxin biosensor uses an engineered tryptophan repressor from *Escherichia coli* for auxin sensing, and the readout is based on fluorescence resonance energy transfer signal (Herud-Sikimic et al., 2021). This biosensor is effective in dynamically probing the actual concentration and location of auxin inside a cell. Further analysis of auxin responses in the epidermis may reveal the coordination of different cell types in epidermal formation.

## 4.4. Cell cycle

Many key transcription factors driving cell fate determination in the leaf epidermis have been identified using forward genetic screens. However, cell fate is not only controlled by transcription factors, but is also associated with specific cell cycle phases (Chen et al., 2015). Almost 30 years ago, a classic study showed that differentiation can be induced during the G1 phase in isolated pluripotent embryonal carcinoma cells, but not at any other cell cycle phase (Mummery et al., 1987). Earlier studies on plant xylem cell differentiation had suggested that DNA synthesis was required (entering the S phase) before xylem cell differentiation, which was contradictory to the G1 phase observations. A model was proposed to resolve this apparent conflict by which cells in early G1 receive a signal and enter differentiation, whereas cells in late G1 need to go through all other phases (S, G2, and M) before reaching early G1 to differentiate (Dodds, 1981). A recent study showed that manipulating the duration of the G1 phase by overexpressing the CDK inhibitor gene *SIAMESE-RELATED4* (*SMR4*) can reduce cell proliferation of the meristemoid (Han et al., 2022).

During stomatal development, SLGCs can serve as a stem-cell-like pool to replenish stomatal precursors or differentiate into pavement cells. According to the transcriptomic signatures of SLGCs, proliferation and differentiation are regulated by both activators and repressors of mitosis and endocycles (Ho et al., 2021), indicating a connection between cell fate and the cell cycle machinery. Low endoreplication levels can result in loss of trichome identity, with cells dedifferentiating into pavement cells (Bramsiepe et al., 2010). When endocycle inhibitors are genetically inactivated, endoreduplicated cells inappropriately evoke a guard cell identity (Iwata et al., 2011). As with the leaf epidermis, the formation of giant cells in Arabidopsis sepals also requires a CDK inhibitor, LOSS OF GIANT CELLS FROM ORGANS (LOG), at the G2-to-M phase transition to produce endoreduplicated cells (Roeder et al., 2010). The ploidy level affects nuclear size, nuclear pore density, chromatin compaction, and the distance between transcription sites and nuclear pores (Robinson et al., 2018; Roeder et al., 2022). These factors may further affect chromatin accessibility and alter the transcriptional profile of cells.

Stomatal patterning is mediated by asymmetric cell division driven by SPCH and symmetric cell division driven by MUTE and FAMA. Using time-lapse imaging, Han et al. (2022) identified a longer cell cycle length in symmetric cell division compared to that of asymmetric cell division. Further analysis showed that the cell fate determinator MUTE directly upregulates the CDK inhibitor gene *SMR4* to slow down the G1 phase and allow the transition to cell differentiation. During Arabidopsis sepal development, the pattern of giant cells is associated with the fluctuating concentration of the transcription factor ATML1 during the G2 phase (Meyer et al., 2017). If ATML1 levels pass a threshold during the G2 phase, the cell will likely enter endoreduplication and become a giant cell. To probe and quantify the length of different cell cycle phases (e.g., G1, S, G2 and M), live-cell imaging tools have been developed such as Cell Cycle Tracking in Plant Cells (Cytrap) (Yin et al., 2014) and plant cell cycle indicator (PlaCCI) (Desvoyes et al., 2020).

Cytrap is a dual-colour system with an S+G2 phase marker and a G2/M phase reporter (Yin et al., 2014). Arabidopsis CDT1a functions in DNA replication origin licencing. Its C-terminal region is responsible for the proteasome-mediated degradation at the late G2 or early M phase. Therefore, the C terminus of CDT1a was fused to the red fluorescent protein (RFP), and the encoding expression cassette was driven by an S-specific promoter from *HISTONE THREE RELATED2* (*HTR2*), a histone 3.1-type gene, to create the S+G2 reporter. Together with the G2/M phase-specific *CYCB1-GFP* marker (encoding a fusion between cyclin B1 and green fluorescent protein), this system allows the visualisation of both S-to-G2 and G2-to-M cell cycle stages. Since Cytrap uses GFP and RFP, the protein of interest can be fused to cyan fluorescent protein (CFP) to trace its dynamics during the cell cycle (Figure 3b).

In PlaCCI, *CDT1a-CFP*, *HTR13-mCherry*, and *CYCB1;1-GFP* are driven by their own promoters to identify G1, S + early G2, and late G2 + M (prophase and metaphase) phase cells, respectively (Desvoyes et al., 2020). All three reporters were integrated into a single transgene, thus facilitating transgenic plant production. Further analysis using these tools to identify different cell cycle phases and the parameters of cell size and shape should improve our understanding of the relationship between the cell cycle and cell fate determination (Figure 3b).

## 4.5. Cell division and cell size

Arabidopsis leaf development is driven by two processes: cell division and cell expansion (Green & Bauer, 1977). Spatiotemporal leaf epidermal development exhibits a proximal zone of division competence and a distal zone with expansion capacity (Fox et al., 2018). Analysing wild-type and *spch* mutant plants revealed the cell-autonomous function of SPCH in promoting division at smaller cell sizes and/or for shorter cell cycle length (Fox et al., 2018). However, the growth rate was similar between wild-type and *spch* plants, suggesting that cell division is uncoupled from cell expansion (Fox et al., 2018). A combination of time-lapse imaging of epidermal growth and modelling in wild-type plants also showed that the average cell cycle duration remained constant throughout epidermal development (Asl et al., 2011). No maximum cell size threshold was found for cell division during pavement cell formation, suggesting that the cell cycle rather than cell size controls cell division events (Asl et al., 2011). In SAM, a study showed that both G1/S and G2/M transitions are size-dependent, with larger cells exiting the G1 phase or entering the subsequent S phase more rapidly than smaller cells, thereby generating uniformly sized cells (Jones et al., 2017). The cell size checkpoint is not triggered by the G2/M phase for cell division in SAM, ruling out the models

that cells undergo cell division at a fixed time after birth or there is a critical size increment between divisions (Willis et al., 2016). Rather, the observation showed that the cell size fluctuations decay by ~75% is intermediate between critical increment and critical size, suggesting a diluter mechanism may account for the cell size regulation (Willis et al., 2016). An asymmetric cell division dilution model then suggests that DNA acts as an internal scale for cell size (D'Ario et al., 2021). In this inhibitor dilution model, one of the cell cycle inhibitors, KRP4 (KIP-RELATED PROTEIN4), is more diluted in the larger cell after cell division. Therefore, the larger cell enters the cell cycle faster than the smaller cell. The observations from SAM formation and leaf epidermal development indicate the importance of cell cycle and asymmetric cell division in controlling cell size homeostasis in the complex environment of multicellular tissues. The SAM is a zone containing division-competent plant stem cells, while the leaf epidermis comprises hundreds of lineages at different stages of cell division and differentiation, thus making the leaf epidermis a more complex system.

In the context of a tissue, topology and geometry are two important aspects to consider, in addition to division rate, when studying the mechanisms underlying epidermal development. Geometry refers to cell shape and size, whereas topology refers to their connectivity within the tissue, for example, the number of neighbours each cell has (Carter et al., 2017). Surprisingly, the topological distribution for the entire cell population is identical between the epidermis of young and old leaves in *spch* plants, which do not have stomata (Carter et al., 2017). This finding suggested that the whole leaf maintains globally topological homoeostasis across developmental time and developmentally distinct zones such as proximal and distal regions (Carter et al., 2017). However, as the epidermis is made of different types of epidermal cells in plants, whether the cell type-specific growth locally shapes a tissue remains unclear. The comparison of the epidermal growth in WT and *spch* cotyledons indicates that stomatal differentiation through the lineage is the driving force for growth variability (Le Gloanec et al., 2022). Those local growth differences are buffered by the adjacent cells of stomata and trichomes to ensure reproducible development (Le Gloanec et al., 2022). Furthermore, to generate diverse leaf shapes, the combination and the arrangement of cell division and cell expansion need to be considered in different zones during leaf development. The molecular interplay between global differentiation across the entire leaf and the local patterning of growing foci along the margin are the keys to making a simple leaf or a dissected leaf in different species (Kierzkowski et al., 2019).

Endoreduplication is a means to increasing DNA abundance in a single cell, and it is often associated with cell differentiation, as mature trichomes and pavement cells are polyploid in the leaf epidermis. Many studies have reported a positive correlation between DNA content and cell size during development (Melaragno et al., 1993; Sugimoto-Shirasu & Roberts, 2003). Therefore, the coordination of endoreduplication and cell size is crucial for organ morphogenesis. But how are these two phenomena related? Robinson et al. (2018) used live-cell imaging and quantitative approaches to investigate the nuclear volume, cell size, cell number, and organ size in sepals of mutants bearing different levels of endopolyploidy or whole-genome ploidy levels (diploid, tetraploid and octoploid). In a perfect compensation model, organ size would remain the same when cell size increases but would be accompanied by fewer cells, such that the area of eight diploid cells would be equal that of two octoploid cells. However, empirical data showed that cell size generally scales up with organ size, although the magnitude of this effect is dampened by compensation, meaning that the cell

size increases with ploidy levels but only to a point (Robinson et al., 2018). The scaling effect is also cell-type specific. For example, the size of pavement cells in sepals increases linearly while the size of sepal guard cells increases exponentially in response to elevated ploidy levels (Robinson et al., 2018). Compared to the inner layers of leaves, the leaf epidermis responds more strongly to changes in ploidy by increasing cell size (Katagiri et al., 2016). This effect may be due to the air spaces in the inner tissue layers beneath the epidermis; to maintain the surface intact, mechanical forces imposed by cell-to-cell adhesion may produce pavement cells of various sizes. Although cell size remains similar in a mutant with smaller nuclei like *crowded nuclei1* (*crwn1*), it is possible that DNA copy number or components from the nucleus or the nuclear envelope sense the change in ploidy and ultimately affect cell size (Robinson et al., 2018). In contrast to sepal development, stomatal formation occurs heterogeneously on the leaf surface and is not synchronised during development, making stomatal density a proxy to probe the termination of cell division (or reduced division events). The observation that sepals with high ploidy levels terminate cell division sooner than sepals with lower ploidy levels is consistent with the reduction in stomatal density as ploidy increases (Robinson et al., 2018).

In summary, with the available tools to obtain quantitative measurements of cell properties and signal concentrations in a spatiotemporal manner, we can start to use modelling approaches to systemically understand the development and predict the output of biological systems, such as patterning and cell fates.

## 5. Future prospects

Leaf epidermal development includes the development of stomatal and trichome cells into a functional tissue in which either stomata or trichomes are well separated from each other. Biochemical signalling, polarity establishment, and growth tension are integrated into the formation of complicated patterns. Although we have a good understanding of stomatal and trichome formation from genetic studies, we still do not understand how these two cell types communicate during development. For instance, we do not understand what determines the differentiation of protoderm cells into stomata, trichomes, or pavement cells or what governs SLGCs to become pavement cells or go through another round of asymmetric cell division.

Development is a continuous process that results in a heterogeneous population. Quantitative biology provides the methods and tools to describe this heterogeneous state. The studies and tools presented here provide the foundation to characterising leaf epidermis phenotypes and quantifying cellular behaviour at a single-cell level. Utilising cell-specific measurements such as polarity level, cell cycle status, transcription factor dynamics, and cell geometry over time can help us link mechanistic descriptions to observed phenotypes. Combined with computational modelling, these cell-specific measurements can help us determine the potential flow of information to illustrate developmental trajectories or even predict cell behaviour. Using simulations, the transcriptional or translational levels of key factors, cell-type specialisation, and pattern formation have been modelled to understand the formation of the SAM and Arabidopsis sepals (Klawe et al., 2020; Meyer et al., 2017).

To better understand leaf epidermal development, we need to further investigate the coordination of different cell types during their development. For example, tracing the dynamics of stomatal and trichome markers in the early developmental stage could help reveal interactions and fluctuations of cell state transitions in

protoderm cells. Although there is no pavement-cell-type-specific marker, quantitative biology can be used to describe the transition from one cell state to another. Phenotypes (e.g., auxin gradients and lobe formation) associated with pavement cell maturation can allow monitoring of transition states to a mature state using time-lapse imaging.

Fueled by the power of single-cell multiomics data, cell identity can be precisely defined by the quantitative and high-resolution integration of gene expression and protein abundance profiles. Recently, numerous studies have investigated transcriptional patterns during leaf epidermal development at single-cell resolution (Xia et al., 2022; Zhang et al., 2021). These data are useful for reconstructing gene regulatory networks and identifying novel genes involved in cell fate commitment (Liu et al., 2020; Lopez-Anido et al., 2021). An analysis of unsupervised clustering of transcriptome and pseudo-time trajectories revealed the flexibility of cell states during leaf epidermis development (Lopez-Anido et al., 2021). Additionally, assigning a cell cycle stage to each cell identity revealed that cell cycle regulators may coordinate with cell differentiation (Lopez-Anido et al., 2021). However, fluctuations in individual cell states may make it difficult to determine cell fate. Indeed, cell state can change over time when facing different internal or external stimuli. A cell can acquire a stable cell identity with a diverse cell state or phenotype (Morris, 2019). Hence, a dynamic system with precise parameters to describe genotype and phenotype is needed to comprehend fate determination.

There is also a need to integrate transdisciplinary resources to shed light on leaf epidermis development. The Plant Cell Atlas is an exciting community resource that integrates spatiotemporal information from the microscale to the nanoscale and describes the dynamic developmental states of plants (Rhee et al., 2019). The quantitative methods and tools summarised here will be useful to delineate epidermal phenotypes and cellular dynamics in cell state transitions. We hope that by incorporating transcriptome, genetic, and signalling network analyses, we will be able to predict cell behaviour and improve our understanding of the fundamental principles governing cell fate and leaf epidermal development.

## Acknowledgements

We thank the reviewers for their advice and Academia Sinica for the support.

**Financial support.** This work was supported by Academia Sinica (grant number AS-CDA-111-L01).

**Conflict of interest.** The authors declare none.

**Authorship contributions.** C.K., S.-L.Y., and C.-M.K.H. wrote the manuscript.

**Data availability statement.** Data sharing is not applicable to this article as no datasets were generated or analysed here.

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
