## [Reviewer Report]

*Comments to Author*: This review written by Chi Kuan, Shao-Li Yang and Chin-Min Ho concisely discusses the formation of Arabidopsis thaliana epidermal cell types and gives a broad summary of quantitative methods used to describe leaf and epidermal cell fate parameters.

This review is well structured into several paragraphs, describing the formation of different epidermal cell types, and then discussing the quantitative methods generally used in leaf studies, here again, the authors focus on the leaf as a whole and the previously described epidermal cell types: pavement cells, stomata and trichomes.

Next, the authors list examples for several selected parameters such as the quantitative measurement of polarity, cell cycle, signals in general and auxin in particular and cell size, all described in their role in cell fate determination of epidermal cells with a focus on measurable parameters.

The discussed topic is summed up in the last part where the authors also give a future prospect on the quickly emerging field.

The text is accompanied by 3 aesthetically pleasing and very concise figures that summarize the key factors guiding the developmental trajectory of the leaf epidermis cell types on a general cell fate as well as a molecular level, illustrate the quantitative methods used to describe leaf morphology and parameters to describe the different epidermal cell types, and describe quantitative tools for measuring polarity and cell cycle. Additionally, all computational tools are listed in a well-structured table, classified by structure analyzed and handily supplied with the publication describing the tool as well as further detailed information on usage.

Overall this review is not only very well written and accompanied by beautiful and comprehensive figures and graphical abstract but can serve as a useful overview of a collection of methods for quantitative analysis of plant epidermal development.

One small general point I would like to address is that in the main part of the text when summarising the biological processes before listing the quantitative methods, the authors could indicate the transition between the two things a bit more distinctly as it is done e.g. in the auxin paragraph but e.g. a bit less in the polarity paragraph.

Specific minor suggestions are listed below that could improve the manuscript:

L76: elaborate or rephrase “to function”

L87: add comma “cells, possessing [...] shape, are associated..”

L261-262: stoma and stomate are used as singular for stomata. Consider using consistent form (suggested: stoma). see also L779, L785

L268: rephrase

279: BASL is also a polarly localized protein. rephrase

L451: typo “used to monitor a transition state”

L453: single-cell omics data

L795 L804: consider keeping the figure legend consistent. in A and B the legend title was a full sentence

L850: the tool is called POME not POEM according to Gong et al. 2021 (see also L870)

---

## [Reviewer Report]

*Comments to Author*: The manuscript by Kuan et al reviews the current understanding of the formation of the Arabidopsis leaf epidermis as well as the quantitative tools that have been used to characterise it. The manuscript is well written, nicely structured and easy to read, and would be of interest to the readership of the Quantitative Plant Biology Journal. Yet, there are some points I would like to comment on, most importantly related to expanding and covering some further relevant literature in the field;

Major:

- Although the authors comment on modelling aspects during the text, there are some other modelling papers that have helped in understanding the regulatory network and patterning aspects in trichomes and in stomata. Even if they are not mentioned it detail, it would be good the readership is aware about this literature. Including some of these papers might require an update of the figures. For instance,

For the trichome side, I would suggest to mention: one of the earlier papers or reviews that mention the Activitor-Inhibitor and Activator-Depletion Model, eg Pesch and Hülskamp 2009 Curr Op Plant Biol, and Balkunde et al 2020 Cell Reports, which is a recent paper proposing how a more updated version of the regulatory network might be operating.

For the stomatal side, it would be also good to mention: Robinson et al 2011 Science, Horst et al 2015 Plos Genetics, and perhaps Han et al 2018 Dev Cell.

I am not necessarily asking to have a modelling section, although the authors could decide to do so, but rather to acknowlegde some of these references given they have been important for understanding cell fate decision making and the regulatory network in the leaf.

- The review is more focused in some parts to stomata, which is totally fine, but in some parts one would expect to have more references to trichomes or more generally to the leaf epidermis. This happens more clearly in the 'signals' section.

-I would suggest to expand more on the cell size and division and include some relevant papers such as Fox et al (2018) Plos Biol, Carter et al (2017) Development, Kierkowski et al (2019) Cell, Kheibarshekan Asl et al (2011) Plant Physiol.

Minor:

• I am aware it is difficult to cover all the relevant literature in this field, so I would suggest to have a disclaimer in the manuscript in which you say you will just cover part of it. If there are relevant specific topics related to quantitative aspects in the leaf epidermis that you are not covering, I would encourage to state it in the manuscript.

• About the patterning analysis, consider to add Naulin et al 2017 New Phytologist, the recent prespective paper by Liu et al (2020) in Front Plant Sci and also other references focusing more on pattern variability such as Greese et al 2014 Frontiers in Plant Sci.

• About the single cell RNAseq, there are two recent references I would ask to mention, Zhang et al 2021 Dev Cell and Xia et al 2022 Dev cell.

• When talking about cell size and ploidy, I would suggest to comment on Robinson et al 2018 The Plant Cell (note that this is in sepal).

• When referring to MGX, it would be better that, instead of pointing to their video tutorial, the MGX 2.0 paper is cited (see Strauss et al 2022 elife). Also, in my understanding, MGX 2.0 can use now deep learning for segmentation as well, but the cell classification as you mentioned is through an SVM classifier, which is a supervised machine learning tool, but does not rely on a neural network (so it would not be deep learning). As explained in Strauss et al, the authors could comment that shape morphology and positional information can be used for cell type classification.

• I would suggest that there is an explanatory caption associated to Table 1.

• PlaCCi is introduced twice, see lines 341 and 352.

• There are some typos, and although the article is nicely written, there are some sentences that might have some grammatical mistakes.

---

## [Reviewer Report]

*Comments to Author*: I appreciate the efforts made by the authors in addressing my comments, the manuscript now has been significantly improved. However, I have some additional minor comments and suggestions I would like the authors to consider. Please find them below:

(The authors might find be a few suggestions that I could have raised in the first round of revision, but I might have missed; if this is the case, I am sorry about it, but I hope such suggestions help in the improvement of the manuscript.)

-Although I appreciate the importance of the first new included section entitled ‘the epidermis as a lod-bearing layer for organ morphogenesis’, I am wondering whether the authors expand too much on it, and the focus of the review, which should be on quantitative methods to understand leaf epidermal development, is a bit more diluted with too many biological details. A possibility to fix it would be to shorten that section (or place part of it elsewhere in the text), such the reader gets more directed to the core of the quantitative aspects in the review.

-Although in the text asymmetric cell divisions in the stomatal lineage are mentioned, perhaps it might be worth emphazising in Section 2.2 that there can be several rounds of such divisions, which will impact on the resulting final pattern (Gong et al 2021, eLife).

-When describing the shape of pavement cells, it would be worth also mentioning the work by Y.E. Sanchez-Corrales (2018) in Development.

-Lines 215-217, please rephrase ““Trichomes can be described using the following parameters: cell proportion on the leaf epidermis, parameters for individual trichome cells, and trichome patterns (distribution).”

One possible rephrasing option is as follows:

“The presence of trichomes in the epidermis, their individual shape and their spatial organization can be described using different sets of parameters.”

-Line 237, please rephrase “trichome patterning is a critical parameter”, given trichome patterning is not a parameter

-Line 242, I suggest to change “using the nearest neighbor distance (NND) (Okamoto et al., 2020).” by

“using the nearest neighbor distance (NND) between trichomes (Okamoto et al., 2020), where each individual trichome is considered as a point, what we will refer as the point-pattern approximation.” so the point-pattern analysis is introduced here and hot in lines 256-258)

-Line 248, I would also add that in the review by Greese at al they review several methods using the point pattern approximation, comparing experimental data and simulations. For consistency with the stomatal section afterwards, it might be good to mention a bit more here the point pattern analysis.

-Line 249, variation-> variability, to be consistent.

-Line 256, “In this point-pattern analysis,” I would add something like “similar to the described analysis in trichomes”, given I suggest the point-pattern is introduced earlier.

-Paragraph starting with line 267: I would suggest to take out the numbers (1), (2), (3), etc. Also, it might be good to expand that paragraph a bit more and comment on stomatal observables found in other publications such as Liu et al.

-Line 282, I would suggest to use another connector instead of ‘in addition to stomatal morphology’, given the previous paragraphs describe more than ‘stomatal morphology’. Note also that durint the text, ‘in addition’ is used many times, so consider using other connectors.

-Line 322, I would suggest to find a smoother transition to the time-laspe tracking section in section 3.5, rather than straightly discribing StomataCounter.

-Line 332, I would rephrase into something like '(…) and PlantSeg (Wolny et al. 2020) allow performing 3D cell segmentation and extract quantitative measures at the single cell level', to avoid confusion, given that PlantSeg does not allow the selection of cells, but its segmentation.

-Paragraph starting with line 438. It might be worth emphasising a bit more that in Zeng et al (2020) they use quantitative measures that provide information at a longer range distances, as opposed to the NND, which provides more local information about the pattern (and/or also you could consider to bring this point up in section 3.3).

-448-449 I would suggest slightly change the start the paragraph, such that it is more connected to the previous one. Perhaps a slight rephrase such as:

“The interplay of cell-to-cell signalling together with the regulatory circuit controlling stomatal initiation is key in generating a pattern, in addition to the polarity-switch mechanism already mentioned”

-In the new section referring to cell division, current models of cell division in the meristem are mentioned, so I would also suggest to mention Willis et al 2016 in PNAS.

-Lines 591-593: I would somehow clarify that the conclusions about the topology are from spch leaves, which do not have stomata.

-It might be interesting to also cite the new paper by Le Gloanec et al 2022 in Development.

-Some typos were detected, eg.:

Fig. 1A -> the ML1 abbreviation in the panel should be ATML1 for consistency.

358 typo GMX2.0 -> MGX2.0

---

## [Reviewer Report]

*Comments to Author*: The revision was nearly sufficient. Some more minor changes were suggested by a reviewer. Please go through them and consider if these changes would further improve the manuscript.

---

## [Reviewer Report]

*Comments to Author*: Thank you for submitting the revised manuscript.

I am pleased to say, it is now accepted for publication - congratulations!